# Relationships among Perceived Control, Safety Attitude, and Safety Performance: A Case Study on Wastewater Treatment Plant Workers

Chung-Fah Huang [1], Yu-Long Tsai [2] and Wen-Hua Lu [1,*]

[1] Department of Civil Engineering, College of Engineering, National Kaohsiung University of Science and Technology, Kaohsiung 80778, Taiwan; jeffrey@nkust.edu.tw

[2] Water Resources Bureau, Kaohsiung City Government, Kaohsiung 80778, Taiwan; yulong.tsai@gmail.com

* Correspondence: dar302@ms78.hinet.net

**Abstract:** Wastewater treatment plants (WWTPs) are an indispensable part of the infrastructure of modern cities. However, because of the existence of many confined working spaces in them, they also pose significant risks of occupational hazards for workers. Therefore, this study was conducted on WWTP workers in Kaohsiung, Taiwan to explore the connections among the perceived control, safety attitude, and safety performance of WWTP workers. In total, 123 valid questionnaires were returned for descriptive statistical analysis, variance analysis, correlation analysis, and hierarchical regression analysis. According to the analysis results, the WWTP workers in this study indicated a mid to high level of perceived control, and they generally believed they were also responsible for industrial safety management. The variance analysis results showed that workers of a different gender, age, service unit, and seniority had significantly different safety attitudes. The hierarchical regression analysis results indicated that the perceived control of the WWTP workers had a significant influence on their safety performance through their safety attitude, which served as a mediator between perceived control and safety performance. It is hoped that these findings can provide references for WWTP managers and workers in their daily communication, operation, and safety management system introduction to ensure better safety.

**Keywords:** perceived control; safety attitude; safety climate; safety performance; hierarchical regression; wastewater treatment plant (WWTP)

## 1. Introduction

Wastewater treatment, a fundamental part of national infrastructure, is considered one of the major indicators of urban modernization. In the IMD World Competitiveness Yearbook published by the Institute for Management Development (IMD) in Lausanne, Switzerland, the percentage of the population served by wastewater treatment plants (WWTPs) is listed as an indicator in the "Health and Environment" dimension for the life quality ranking. Better wastewater treatment can help to improve not only a country's image and competitiveness but also the quality of its environment by collecting, treating, and finally releasing wastewater into the environment when the wastewater meets the environmental standards.

Though different in equipment type and scale, WWTPs share the typical risks of operational hazards for workers, such as falling, tumbling, getting caught in equipment, getting hit by falling objects, electric shocks, fire, explosion, drowning, and hypoxia. In the working spaces of the plants, such as sewage lagoons and tanks, major incidents of workers being injured or killed by hypoxia, hydrogen sulfide (H2S) intoxication, electric shocks, and explosion have been reported [1]. According to the post hoc investigation findings of these incidents, the causes are mostly the failure of workers to follow safety protocols and the failure of managers to supervise, inspect, give employees required safety training, and promote safety awareness.



Though different in equipment type and scale for different amounts of wastewater to treat, WWTPs are generally composed of semi-closed pools and tanks equipped with electric machines, such as pumps, screeners, sludge scrapers, dehydrators, and conveyors, for different steps of wastewater treatment. To ensure the proper operation of the machines, maintenance and repair are frequently needed, requiring workers to dismantle the machines for the removal of sludge and rust, go into the tanks, and/or move heavy objects. Under such circumstances, WWTP workers are facing the risks of the typical occupational hazards at other plants, such as falling, tumbling, getting caught in equipment, getting hit by falling objects, electric shocks, explosion, fire, and drowning. According to the Occupational Safety and Health Act in Taiwan, the risks of occupational hazards at WWTPs are ranked as the most significant risks. Therefore, it is very important to strictly follow the safety management and operation rules at WWTPs, particularly in semi-confined and confined working spaces, such as the sewage pools and tanks. In confined working spaces, there may be particular physical hazards, such as objects falling on workers, pipelines broken under high temperature and pressure, unprotected machinery, and electrical shocks caused by exposed wires [1]. According to an information sheet issued by the NZ Department of Labour, "It's been calculated that working in a confined space is 150 times more dangerous than doing the same job outside" [2,3].

It is of great importance to study the safety self-control, safety-awareness, safety attitude, and observance of safety protocols of the WWTP workers in such a hazardous working environment as these factors will influence the operational safety of the water treatment plant as a whole. However, research on the perceived control, safety attitude, and safety performance of WWTP workers is significantly lacking. Therefore, this study was conducted to explore the influence of WWTP workers' perceived control and safety attitude on their safety performance. Through a review of both domestic and international literature combined with the consideration of the operation and management of WWTPs in Taiwan, a questionnaire was developed for the survey of this study on WWTP workers in Kaohsiung to explore the interconnections among the perceived control, safety attitude, and safety performance and also to provide practical suggestions for WWTPs based on the analysis of the findings.

## 2. Literature Review

### 2.1. Operation and Management of WWTPs and Occupational Hazards in Confined Spaces

In many different industries, there are a wide variety of confined working spaces, such as sewers, wastewater tanks, shafts, and tower tanks; moreover, these confined working spaces are generally not regular work sites [1]. As found in many studies, deaths in confined working spaces are significantly caused by physical hazards, such as falls, electrocution, and being caught or crushed in machines [4–7]. According to the statistics from 2010 to 2019, as illustrated in Figures 1 and 2, there were 59 deaths of workers caused by occupational hazards in confined working spaces in Taiwan. Most of them occurred in storage tanks, sewers, and manholes, followed by wastewater pools (tanks). Twenty-nine of the deaths were caused by H2S intoxication, followed by 17 deaths caused by hypoxia [8]. H2S is a kind of colorless gas with a unique rotten-egg smell. It will cause temporary loss of smell at a concentration of 100 ppm, dizziness and nausea at 200 ppm, and death at 300 ppm or higher. It often exists naturally in crude oil, natural gas, and hot springs. It can also be generated by the bacterial decomposition of organic matter. Therefore, places such as wastewater treatment plants, pastures, and excrement storage tanks tend to have higher H2S concentrations and lower oxygen levels. When the oxygen level is lower than 18%, people will suffer from hypoxia, which will result in dyspnea, dizziness, and even death. If workers do not have sufficient safety awareness and implement working environmental safety rules, this toxic gas can easily cause dyspnea or even death for it cannot be detected by the human nose. Research on deaths in confined working spaces also indicates incidents in which rescuers were killed or injured—most of whom were not professional rescue workers but workmates who tried to save their coworkers [2,9,10].

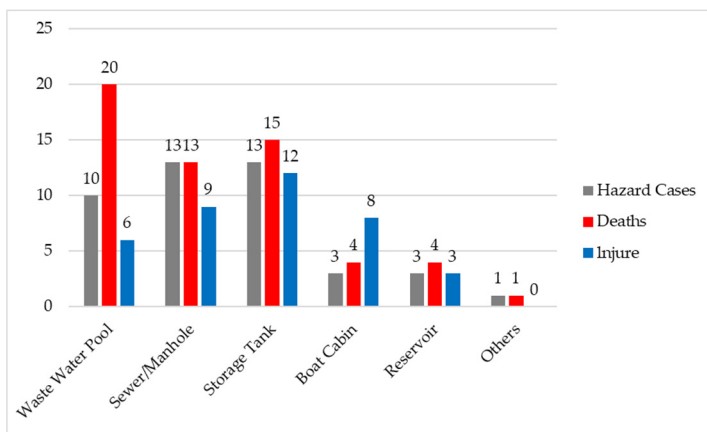

**Figure 1.** Statistics of Hypoxia in Different Confined Working Spaces in Taiwan from 2011 to 2020.

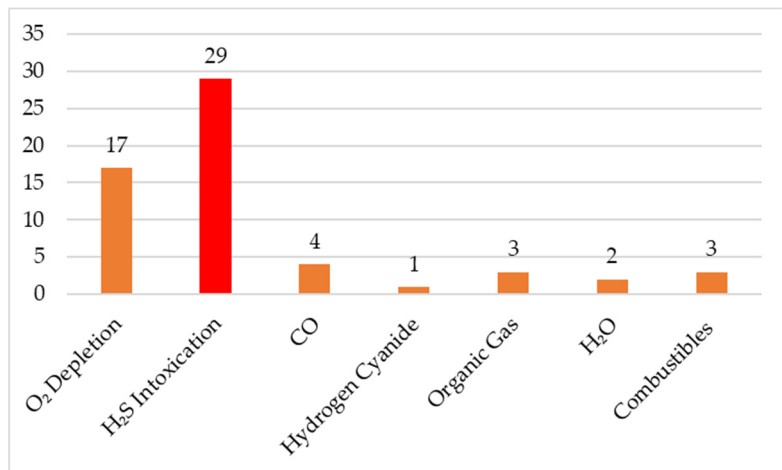

**Figure 2.** Statistics of Deaths of Workers Caused by Different Media in Confined Working Spaces in Taiwan from 2011 to 2020.

Taiwan has put in place stringent laws and regulations on working environment safety to prevent occupational hazards. For example, for the prevention of dyspnea and toxic gases, Article 29-4 of the Regulations for the Occupational Safety and Health Equipment and Measures stipulates, "When asking workers to work in confined spaces, employers shall prepare equipment to measure/detect the oxygen level and the concentration of hazardous substances within such spaces before and throughout the working process to prevent hazards caused by oxygen deficiency and/or hazardous substances." Articles 311 and 312 of the same act also stipulate the use of ventilation and air exchange equipment capable of providing fresh air, regulating the temperature, reducing concentrations of hazardous substances, and facilitating sufficient air exchange.

Research on industrial accidents first started in the 1930s with the Domino Theory proposed by Heinrich, a pioneer in the field of industrial safety in the US. According to the theory, an accident is the inevitable result of a chain of events that have occurred in a specific and logical sequence. The theory lists five sequential factors of an accident: (1) ancestry and social environment, such as personality faults caused by heredity and/or the social environment; (2) individual's mistake, including congenital or acquired faults, such as carelessness, irritability, and negligence of operational safety; (3) unsafe action and/or physical hazard, including inappropriate behaviors and/or unprotected mechanical/physical operations resulting in hazardous accidents; (4) actual accident, such as falling, tripping, or crashing; and (5) injury from the accident, such as death, bone fracture, or laceration. The elimination of individuals' mistakes and avoidance of unsafe actions can result in an immediate improvement in the safety of the working environment. According

to the findings of Heinrich's further research on the causes of industrial accidents, 88% result from unsafe behaviors, 10% from unsafe environments, and only 2% are inevitable. There are already studies that have found various influence factors of safety behaviors and outcomes [11] and explored the influence of the organizational safety climate on safety behaviors and outcomes [12].

The unsafe attitude of workers accounts for a large majority of workplace accidents [13]. According to existing studies, the more positive safety attitude of an organization can lead to higher productivity and fewer accidents [14]. As demonstrated by Smith & Wadsworth [15], there is a strong connection between safety attitude and safety performance.

The influences of the personal factors of WWTP workers, such as their perceived control and safety attitude, on their safety performance are still understudied. In high-risk working environments, such as the confined working spaces at WWTPs, workers' self-control, personal protection measures, and attitude toward work safety and the safety management system could play a decisive role in the safety performance of WWTPs. Therefore, this study was conducted focusing on the personal psychological aspects of workers in order to provide helpful suggestions based on its findings for WWTPs.

### 2.2. Perceived Control

Perceived control is an important research topic in psychology. It has a stronger influence on an individual's decisions, actions, and emotions than actual control. The concept of "effectance motivation" proposed by White & Lippitt [16] refers to the pervasive and intrinsic need of an individual to control the external environment when feeling motivated and capable of doing so. An individual's behaviors are guided by his/her intrinsic motivation, which is subject to the influences of perceived self-control and perceived self-capability. If people perceive that they have more control of the situation and more capability of doing something, they will have higher intrinsic motivation to do it and vice versa. Perceived control refers to one's perceived ability to change or control the condition or environment where they are situated [17]. Empirically speaking, it refers to the level of one's perceived ability to achieve desirable results and/or avoid undesirable results by making efforts to do so. The feelings of perceived control, mastery, and self-efficacy can bring many positive consequences, such as better emotional stability, resilience against stress, health behaviors, occupational safety, and commitment to work. According to the definition by Ajzen [18,19], perceived control refers to one's perception of the difficulty in implementing a behavior. This concept is derived from the concept of perceived behavioral control proposed earlier by Ajzen [18]. Trafimow et al. [20] indicated that perceived behavioral control is composed of perceived control and perceived difficulty. Perceived control refers to the degree to which people believe that they have voluntary control over a certain behavior. Yu et al. [21] pointed out that perceived control is the physiological capability of perceiving the feasibility of achieving expected behaviors and/or performances. Without the sense of perceived control, an individual will not be able to manage work properly. There are currently few studies on the role of perceived control in a work safety setting.

According to the findings of the study by Ito & Brotheridge [22], people with higher perceived control often take the initiative of searching for, learning, and applying work-related information and have a better understanding of the safety standards required by their organizations, which helps to not only effectively reduce role ambiguity but also prevent unsafe behaviors and accidents. Neal & Griffin [23] believed that people with higher perceived control generally tend to believe in their ideas and their own influence on what happens in their life regardless of real-life constrictions. The empirical studies by researchers [23,24] all demonstrate a significant relationship between personal safety behaviors and occupational accidents, indicating that adhesion to the safety protocols can help to reduce the hazards caused by inappropriate operation or behaviors. Neal & Griffin [23] and Spector et al. [25] found that employees with higher perceived control are more capable of controlling both the consequences of their work and changes in the

environment without uncertainty or doubt about their own ability to achieve their targets. This is conducive to the improvement of safety performance. With higher perceived control over their safety behaviors and their observation of safety protocols, workers can effectively reduce the hazards caused by inappropriate operation. Kuo et al. [26] argued that employees with higher perceived control of safety believe safety is controllable rather than preordained and they will take the initiative of meeting the requirements of safety performance to protect their own property and safety. On the other hand, employees with lower perceived control believe safety is something beyond their control and consequently react passively to safety requirements at work for they think accidents are inevitable despite their efforts to ensure safety. Such a passive perception of workers toward work safety tends to result in accidents. Considering the above statements, if perceived control theory is applied to the field of organizational safety research, we can hypothesize that the higher the perception of safety control will be, the more likely workers will be to take the initiative to improve safe behavior.

### 2.3. Safety Attitude

According to the narrow definition proposed by the International Nuclear Safety Advisory Group (INSAG) [27], "safety culture is that assembly of characteristics and attitudes in organizations and individuals which establishes that, as an overriding priority, nuclear plant safety issues receive the attention warranted by their significance." There are many dimensions of safety culture. According to Donald & Canter [28], the safety attitude of employees mean the safety culture of an organization. A questionnaire survey on the safety attitude of employees is helpful for the prediction of accident occurrences and the development of responding measures for safety improvement. The research scope of safety culture is rather wide, while the safety attitude of members in an organization plays a critical role in the formation of its safety culture. Therefore, safety attitude was chosen as one of the dimensions in this study, and it was defined as an employee's or member's attitudes toward safety culture.

Heinrich [29] believed that an inappropriate attitude is one of the personal factors of unsafe behaviors, and one's attitude and awareness have an influence on his or her behaviors and propensity to accidents. Reece & Gable [30] divided attitude into cognitive, behavioral, and affective components. An attitude is a person's or an entity's evaluation of another person, entity, or idea that has a direct influence on the attitude-holder's social behaviors [30,31]. Champoux [32] argues that attitude is composed of not only cognitive and affective components but also behavioral intentions: the intentions of an individual to behave or take actions according to his/her attitude toward an object or issue.

According to the definition by Hannaford [33], attitude is "a readiness to respond effectively and safely, particularly in tension-producing situations". Since back in the 1960s, the study of attitudes has progressed considerably but was then considered as an uninfluential and weak predictor of behaviors [31]. However, since then, the study of attitude has developed significantly and established attitude as an important behavioral predictor, especially for safety behaviors [28]. For example, it was found that attitudes towards safe driving play an important role in promoting safe driving behaviors [34,35]. To solve problems of industrial safety, it is a priority to improve safety attitude and understanding.

### 2.4. Safety Performance

The concept of safety performance centers around the maintenance of workplace safety. It can be achieved through the worker safety behaviors under the organizational safety culture or the ability to prevent accidents and occupational injuries on work sites [36,37]. According to Petersen et al. [38], the biggest challenge with safety is safety performance measurement. How supervisors implement safety inspection and safety performance measurement reflects how committed they are to ensuring and improving the safety performance. Such commitment is an indispensable part of the promotion of safety culture. The measurement of certain aspects of safety performance, such as accident information or

data, can be used as a direct indicator of occupational safety performance. However, it will take some time to collect and verify such information or data. No history of occupational hazards or accidents is necessarily equal to good safety performance. Through measurement methods such as the safety performance indicators, it is possible to understand if resources invested in safety performance deliver intended results.

Many studies have found that safety performance enhancement can be either proactive or reactive. For example, the British Standard Institution (BSI) adopted both proactive and reactive indicators for the evaluation of labor health and safety system performance and enhancement. Reactive safety performance enhancement focuses more on reducing injury frequency rates and compensation costs [39] caused by incidents, such as unsafe behaviors, unsafe conditions, false alarms, and/or incidents that cause only financial losses. Reactive safety performance enhancement evaluates safety performance based on historical data other than the status quo [40]. Proactive safety performance enhancement focuses on evaluating the safety climate, safety culture, hazard identification, and observation of a region, organization, or entity. It may take a long period of time for the consequences of many occupational hazards to emerge. Therefore, the measures of correction or improvement taken after the emergence of hazard consequences can be too late. Proactive safety performance measurement and enhancement is more preemptive. It is also intended to incentivize organizations or entities to detect and improve what can be improved.

Burke et al. [41] conducted a study on 550 toxic waste disposal officers and developed their General Safety-Performance scales with four dimensions of safety performance through a factor analysis: "Using Personal Protective Equipment", "Engaging in Work Practices to Reduce Risk", "Communicating Health and Safety Information", and "Exercising Employee Rights and Responsibilities". Snyder et al. [42] used the three dimensions for safety performance measurement and found that safety understanding and supervisor support had a direct influence on safety performance through safety control.

There are already many studies on safety training, equipment safety, and accident investigation as influence factors of safety performance, with a majority of them focusing particularly on safety training and education. For example, in the literature on construction safety, there are numerous studies focusing on safety training as an important factor in safety improvement [43].

It has been noted that there is a shift of focus from organizational accidents [44] to the role of the safety climate for the prediction and prevention of workplace accidents and injuries [45]. As the safety climate serves as a frame of reference for the behaviors and attitudes of the employees, it also has an influence on their accident involvement. With more favorable safety perceptions (reflecting a more positive safety climate), workers are less likely to have unsafe acts [46] and consequently have less accidents and injuries [47]. However, the perceptions of the safety climate among workers may vary among different industries and regions [48].

Among the several methods to improve the safety performance of construction organizations, one is the safety climate approach [49]. Umar & Umeokafor [50] also suggested that one of the approaches to improve the safety performance of construction organizations is the safety climate approach, which is helpful in measuring how mature the safety climate of an organization is and then developing a plan to reach the desired level of maturity. Seeing as there are few safety climate measurement tools developed particularly for the construction industry, they developed a safety climate assessment tool for the Gulf Cooperation Council construction industry through email interviews held with construction industry professionals. It is a new assessment tool that has seven factors; however, longitudinal studies are needed to further establish the effectiveness of the tool [50].

However, such personal psychological dimensions of WWTP workers have been understudied. This study is particularly meaningful with its exploration of such dimensions to provide more information for better WWTP safety management. In this study, the safety climate of a WWTP perceived by its workers was used as the proactive safety performance measurement indicator in its design of questionnaires.

## 3. Research Methods

### 3.1. Research Hypotheses

According to the above-mentioned review of literature, lower perceived control of workers will result in negative results and negative influences on work performance and safety behaviors, while better safety attitude of workers stimulated by their working environments and relationships with their co-workers will result in better safety behaviors and, consequently, better safety performance.

As found in the study by Kuo et al. [51], perceived control is a significant predictor of safety behavior climate and occupational accidents through safety culture; the safety attitude as a sub-dimension of safety culture was selected in this study to test the influence of safety attitude of WWTP workers as a mediator. With WWTP workers' perceived control as the independent variable, safety performance as the dependent variable, and safety attitude as the mediator variable, the following hypotheses were developed in this study:

**Hypothesis 1**: *Perceived control of WWTP workers has a direct positive influence on their safety attitude.*

**Hypothesis 2**: *Perceived control of WWTP workers has a direct positive influence on their safety performance.*

**Hypothesis 3**: *Safety attitude of WWTP workers has a direct positive influence on their safety performance.*

**Hypothesis 4**: *Safety attitude of WWTP workers has a mediation effect between their perceived control and safety performance.*

### 3.2. Questionnaire Design and Sampling Method

The models and hypotheses of this study were established to explore the relationships among the three major variables: perceived control, safety attitude, and safety performance of WWTP workers. A questionnaire was designed in this study with questions covering the dimensions of these three variables. The references used to develop the questions of each dimension are explained as follows.

For the perceived control dimension, the definition of perceived control proposed by Kraft et al. [51] was used in this study. According to this definition, "perceived control" refers to one's perceived level of control over the consequences of his/her behaviors. Related studies also support that lower perceived control will result in negative consequences and consequently affect work performance and safety behaviors. The scale of perceived control developed by Kuo et al. [26] in their long-term empirical research on labor safety and health has a high level of reliability (with an internal consistency coefficient of 0.78). In this study, references were drawn from the research by Kraft et al. [51] and Kuo et al. [26] to develop, in total, five questions for the perceived control dimension. The five questions were based on a reverse-item design.

The safety attitude in this study refers to workers' attitude toward safety behaviors produced by their feelings about work and safety understanding together with simulation from their work environment and co-workers. It was composed of four sub-dimensions in this study: personal safety, co-worker safety, supervisor support, and safety understanding. References were drawn from the viewpoints and scales of Guastello [52], Cranny et al. [53], and Smith et al. [54] to develop, in total, 19 questions for the safety attitude dimension. The safety performance dimension was divided into predictive safety and perceived safety sub-dimensions. References were drawn mainly from the scale developed by Burke et al. [41] and Wu & Kang [55] to develop, in total, 11 questions for the safety performance dimension.

The subjects in this study were workers at four wastewater treatment plants in Kaohsiung, including WWTP workers (including managers, supervisors, and employees) and workers of WWTP operation contractors and maintenance/repair contractors. WWTPs were targeted in this study mainly for two reasons. First, they are workplaces with unique working environments and relatively high risks of occupational hazards. Second, they

are fixed workplaces where workers regularly engage in a limited variety of operation activities. All the employees of the WWTPs and contractors had access to the questionnaire survey of this study. To ensure the accuracy and validity of the questionnaire survey, a pre-test was conducted in this study on ten employees who have been working for over ten years at the four WTTPs and their contractors. Their understanding of the questions and their feedback were collected for the revision of the questionnaires. The finalized questionnaires were distributed and collected by designated personnel of the study team. In total, 160 questionnaires were distributed to and collected from the subjects by designated personnel of this study to motivate more willingness of the subjects to respond. In total, 150 questionnaires (93.7%) were returned. After the exclusion of invalid questionnaires, such as incompletely answered questionnaires, there were 123 valid samples in total (with a valid return rate of 82%).

All the questions were based on a five-point Likert scale, with one point for "Strongly disagree", two points for "Disagree", three points for "Neutral", four points for "Agree", and five points for "Strongly agree". The questionnaire results were collected and analyzed using SPSS for descriptive statistical analysis, ANOVA, correlation analysis, and hierarchical regression analysis.

As indicated in Table 1, which lists the descriptive statistical analysis results of subject profile, 109 of the subjects were male (88.6%) and 14 were female (11.4%). The average age of the subjects was 41.54 years, with the 30–39 age group being the largest (29.3%), followed by the 40–49 age group (27.6%). A majority (43.1%) of the subjects graduated from junior colleges, followed by the subjects who graduated from universities (17.9%). Most of the subjects (77.9%) majored in engineering, including civil engineering, architecture, environmental engineering, mechanic/electrical engineering, and others. A majority of the subjects (40.2%) were employed by the WWTPs (as civil servants for WWTPs in Taiwan are governmental entities), followed by the subjects employed by WWTP operation contractors (40.2%), and by maintenance/repair contractors (18%). In total, 74 subjects (60.7%) were on-site maintenance/repair workers or operators (technicians), and 32 subjects (26.2) were office workers (such as administrative assistants, engineering assistants, and office managers). The average years of working experience among the subjects was 11.27 years (SD = 9.847). A majority of the subjects (47.2%) had five years or less of working experience, followed by 16 to 25 years (31.7%). The subjects, on average, have been working for their current employers for 8.42 years (SD = 9.410). Most of the subjects (62.6%) had been working for their current employers for five years or less, followed by 16 to 25 years (22%).

**Table 1.** Descriptive statistical analysis results of the subject profile.

| Attribute | Classification | No. | Ratio (%) |
|---|---|---|---|
| Gender | Male | 109 | 88.6 |
| | Female | 14 | 11.4 |
| Age | <30 | 24 | 19.5 |
| | 30 to 39 | 36 | 29.3 |
| | 40 to 49 | 34 | 27.6 |
| | ≥50 | 29 | 23.6 |
| Educational Level | High School/Vocational High School | 14 | 11.4 |
| | Junior College | 53 | 43.1 |
| | Technical/Vocational College | 21 | 17.1 |
| | University | 22 | 17.9 |
| | Graduate School or Higher | 13 | 10.6 |
| Major | Engineering (Civil Engineering, Architecture, Hydraulic, Environmental, Mechanic/ Electrical Engineering, etc.) | 95 | 77.9 |
| | Others | 27 | 22.1 |
| Employer | Civil Servants for WWTP | 51 | 41.8 |
| | WWTP Operation Contractor | 49 | 40.2 |
| | Maintenance & Repair Contractor | 22 | 18.0 |

**Table 1.** *Cont.*

| Attribute | Classification | No. | Ratio (%) |
|---|---|---|---|
| Payroll | Full-time Employee | 91 | 76.5 |
| | Contract Employee | 28 | 23.5 |
| Work Position | On-site Maintenance/Repair Worker or Operator (Technician) | 74 | 60.7 |
| | Office Worker (Administrative Assistant, Engineering Assistant, Office Manager, etc.) | 32 | 26.2 |
| | Supervisor (Section Chief, Team Chief, Site Manager, etc.) | 16 | 13.1 |
| Total Years of Working Experience | ≤5 Years | 58 | 47.2 |
| | 6 to 15 Years | 17 | 13.8 |
| | 16 to 25 Years | 39 | 31.7 |
| | >25 Years | 9 | 7.3 |
| Total Years for the Current Employer | ≤5 Years | 77 | 62.6 |
| | 6 to 15 Years | 12 | 9.8 |
| | 16 to 25 Years | 27 | 22.0 |

*3.3. Data Analysis Methodology*

After the exclusion of invalid questionnaires, such as incompletely answered questionnaires, the valid questionnaires were analyzed for descriptive statistical analysis to better understand the current conditions of the WWTP workers in this study. In addition, a *t*-test and an ANOVA test were conducted to explore the connections between different subject profile variables (gender, age, position, and years of working experience) and the dimension variables of this study. Finally, a correlation analysis and a hierarchical regression analysis were conducted to explore the interconnections among the factors.

**4. Results**

The Cronbach's $\alpha$ analysis, a common method of consistency analysis, was used in this study to measure the reliability of the questions for the dimensions and sub-dimensions. If Cronbach's $\alpha$ is <0.35, it means low reliability, 0.35 < Cronbach's $\alpha$ to <0.70 means moderate reliability, and Cronbach's $\alpha$ > 0.70 means high reliability. The analysis results were listed in Table 2.

**Table 2.** Reliability analysis results of the dimensions.

| Dimension | | Cronbach's $\alpha$ |
|---|---|---|
| Perceived Control | | 0.686 |
| Safety Attitude | Personal Safety | 0.591 |
| | Co-worker Safety | 0.851 |
| | Supervisor Support | 0.755 |
| | Safety Understanding | 0.614 |
| Safety Performance | Predictive Safety | 0.877 |
| | Perceived safety | 0.934 |

Among the dimensions of this study, safety attitude was composed of four sub-dimensions and safety performance of two sub-dimensions, while there was no sub-dimension for perceived control. The Cronbach's $\alpha$ values of the co-worker safety sub-dimension ($\alpha$ = 0.851), supervisor support sub-dimension ($\alpha$ = 0.755), predictive safety sub-dimension ($\alpha$ = 0.877), and perceived safety sub-dimension ($\alpha$ = 0.934) were all higher than 0.70, indicating high reliability. The Cronbach's $\alpha$ values of the perceived control dimension ($\alpha$ = 0.686), personal safety sub-dimension ($\alpha$ = 0.591), and safety understanding ($\alpha$ = 0.614) were all higher than 0.5 and lower than 0.70, indicating acceptable reliability levels.

*4.1. Descriptive Statistical Analysis*

The perceived control scale developed in this study took on the reverse-item design as the reference scales. Therefore, higher average scores (with five points as the highest score) on the perceived control scale of this study represent lower perceived control and vice

versa. The average score of each question for the perceived control were listed and ranked in Table 3. The average score of the overall dimension was 1.74 (SD = 0.505), indicating relatively high perceived control among the subjects. The average score of "Work safety management is the responsibility of the safety and health department of the company and has nothing to do with me" was the lowest at 1.43 (SD = 0.544), indicating the subjects generally had positive thoughts and high recognition of their personal participation in work safety management.

**Table 3.** Analysis results of the perceived control dimension.

| Items | Average Score | S.D. |
|---|---|---|
| I believe that nothing will go wrong at work as long as I follow my own judgement. * | 2.04 | 0.918 |
| I believe accidents on the work site are often caused by issues with devices and/or equipment. * | 1.93 | 0.773 |
| I believe that skipping some steps of the standard work procedure will not actually affect work safety. * | 1.73 | 0.747 |
| I feel that accidents will never happen to me. * | 1.55 | 0.749 |
| Work safety management is the responsibility of the safety and health department of the company and has nothing to do with me. * | 1.43 | 0.544 |
| Overall Dimension | 1.74 | 0.505 |

Note: * reverse item; S.D.: standard deviation.

There were 19 questions in total for the four sub-dimensions of the safety attitude dimension in this study. The average scores of these questions were ranked and listed according to the corresponding sub-dimensions in Table 4. The average score of the dimension was 4.18 (SD = 0.388), indicating that the WWTP workers took their own and their co-workers' safety seriously, while their supervisors were proactive in improving the work environment, providing safety training, taking care of the perceptions of their subordinates, and assigning a suitable amount of work. This was particularly reflected in the sub-dimension of safety understanding. The average scores of the four sub-dimensions were, respectively, 4.04 (personal safety), 4.35 (co-worker safety), 4.14 (supervisor support), and 4.18 (safety understanding). Among them, the average score of co-worker safety was the highest, indicating that the co-workers of the subjects generally not only accepted but also took the initiative of giving reminders about worker safety to one another.

There were 11 questions for the safety performance dimension, which are composed of the predictive safety and perceived safety sub-dimensions. The average scores of these questions were ranked and listed in Table 5. The overall average score of the dimension was 4.09 (SD = 0.496), indicating the subjects were positive about the current safety performance of the WWTPs where they were working. While the average score of the perceived safety sub-dimension was 4.25, the average score of predictive safety was only 3.36. Between the two questions for the predictive safety, "I believe that my company will not have any death caused by occupational hazards in the coming year with its current equipment and environment" had a higher average score of 3.38 (SD = 1.218), and "I believe that my company will not have any false alarm incident with its current equipment and environment on the work site" had an average score of 3.34 (SD = 1.214), indicating the subjects generally did not have high recognition of or confidence in the safety protection at their WWTP work sites. This finding is worthy of further attention from WWTP managers to improve the safety management and protection in order to promote higher predictive safety and recognition among WWTP workers.

**Table 4.** Analysis of the safety attitude dimension.

| Sub-Dimension | Items | Average Score | S.D. | Average Score of Sub-Dimension |
|---|---|---|---|---|
| Personal Safety | I believe wearing safety shoes and helmet can help to prevent occupational hazards. | 4.34 | 0.876 | 4.04 |
| | I frequently keep passageways on the work site clean and free of obstruction. | 4.33 | 0.719 | |
| | To ensure safety, I thoroughly understand how to use the protective gear properly and wear it all the time on the work site. | 4.30 | 0.852 | |
| | I believe that the safety rules of the company will reduce my work efficiency. * | 3.75 | 1.083 | |
| | I believe that it is highly hazardous to work on the work site. | 3.44 | 1.160 | |
| Co-worker Safety | I am very willing to accept reminders about work safety from colleagues. | 4.48 | 0.605 | 4.35 |
| | I pay attention to the safety of my colleagues or other workers on the work site and remind them to follow safety rules. | 4.38 | 0.621 | |
| | My colleagues pay attention to on-site safety and remind me of following safety rules. | 4.20 | 0.746 | |
| Supervisor Support | My direct supervisor make frequent round checks and, when finding any hazardous action taken by my colleagues or employees of contractors, will immediately stop it. | 4.24 | 0.728 | 4.14 |
| | My direct supervisor will give warnings to those employees who continue to violate safety rules or have unsafe behaviors after several times of exhortation against such actions. | 4.17 | 0.721 | |
| | When a worker is in poor physical condition, my director supervisor will immediately stop him/her from work. | 4.07 | 0.784 | |
| | My director supervisor frequently check if any of his subordinates has a slack work attitude. | 4.06 | 0.785 | |
| Safety Understanding | A safe working environment can bring better work performance. | 4.48 | 0.578 | 4.18 |
| | If employees take participation in safety training seriously, it is helpful to improve their safety performance. | 4.42 | 0.627 | |
| | Paying attention to how employees feel about their work can help to improve their work performance. | 4.41 | 0.699 | |
| | Providing sufficient safety training can help to reduce the occurrence of accidents. | 4.39 | 0.754 | |
| | Accidents occur to employees mostly because of bad luck. * | 4.25 | 0.911 | |
| | If an employee pays too much attention to the safety procedure, it will impair his/her work efficiency. * | 3.75 | 1.083 | |
| | When an employee is too busy at work, he/she will become negligent about work safety. | 3.57 | 1.160 | |
| | Average Score of the Dimension = 4.18; S.D. of the Dimension = 0.388. | | | |

Note: * reverse item; SD: standard deviation.

According to the *t*-test results of the connections between the subjects' gender and their variances in each dimension, a significant difference was found between the male and female subjects in their perceived control ($t = -2.450$ *, $p < 0.05$). The average score of the female subjects in perceived control was significantly higher than that of the male subjects. It was probably because the female subjects mostly worked in the office and, therefore, were less exposed to hazards on the work site.

According to the ANOVA test results, it was found that subjects of different age groups were significantly different in the safety performance dimension and the sub-dimensions of predictive safety and perceived safety. The average scores of the subjects less than 30 years old in safety performance, predictive safety, and perceived safety were significantly higher than those of the subjects over 50 years old. It was because these younger subjects were mostly employees of WWTP operation contractors with less experience with occupational hazards and, therefore, relatively optimistic predictive safety. Moreover, the average scores of the subjects from different employers were also significantly different in safety performance, predictive safety, and perceived safety. The average scores of the subjects from WWTP operation contractors (including BOT contractors) in safety performance, predictive safety, and perceived safety were significantly higher than the subjects employed by WWTPs (who are essentially civil servants). Finally, in the predictive safety sub-dimension, the average score of the subjects from WWTP operation contractors (including BOT con-

tractors) was significantly higher than that of the subjects from WWTP maintenance and repair contractors.

**Table 5.** Analysis results of the safety performance dimension.

| Sub-Dimension | Items | Average Score | S.D. | Average Score of Sub-Dimension |
|---|---|---|---|---|
| Predictive Safety | I believe that my company will not have any death caused by occupational hazards in the coming year with its current equipment and environment. | 3.38 | 1.218 | 3.36 |
| | I believe that my company will not have any false alarm incident with its current equipment and environment on the work site. | 3.34 | 1.214 | |
| Perceived Safety | All the employees at my company have received general work safety training. | 4.38 | 0.659 | 4.25 |
| | I frequently pay attention to if there is sufficient light, illumination, and ventilation on the work site. | 4.36 | 0.703 | |
| | After each accident, the company will not only have an in-depth investigation about its causes but also announce and compile the investigation findings as materials for safety training. | 4.30 | 0.652 | |
| | The safety and health training of the company meet the safety requirements and I can put what I learn from the training into practice. | 4.27 | 0.690 | |
| | For newly purchased personal protective equipment, the company gives training on how to properly use it. | 4.23 | 0.758 | |
| | I frequently receive reminders of work safety awareness promotion and information about work safety cases from the company. | 4.22 | 0.621 | |
| | The passageways at my work site is clear of obstruction and all the materials and objects are placed in a clean and tidy fashion. | 4.21 | 0.763 | |
| | I strictly implement the measures of access control for contractors' employees on the work site. | 4.20 | 0.709 | |
| | I believe that the managers of the company frequently analyze work hazards to improve work safety. | 4.13 | 0.724 | |
| | Average Score of the Dimension = 4.09; SD of the Dimension = 0.496 | | | |

### 4.2. Correlation Analysis

A Pearson correlation analysis was conducted in this study to explore the correlations between the two variables of the subject profile (age and total years of working experience) and the variables of the dimensions and sub-dimensions of perceived control, safety attitude, and safety performance. The average scores of the subjects in the dimensions and sub-dimensions were used in the correlation analysis.

As several questions in the questionnaire of this study were negative items, the scores of these negative-item questions were reversed to facilitate the analysis and discussion. As indicated in Table 6 that lists the Pearson correlation analysis results, the age and total years of working experience of the subjects were significantly correlated with only some of the sub-dimensions. Nevertheless, the dimensions and sub-dimensions all demonstrated low to mid correlations with one another. In particular, the correlation coefficients between perceived control and the two sub-dimensions of safety attitude, co-worker safety, and safety understanding were the highest, respectively, at r = 0.395, $p < 0.01$, and r = 0.254, $p < 0.01$, indicating low positive correlations. The perceived safety sub-dimension of safety performance was significantly correlated with the safety attitude dimension. All the sub-dimensions of safety attitude were significantly and positively correlated with the perceived safety sub-dimension of safety performance with correlation coefficients, respectively, at r = 0.336, $p < 0.05$; r = 0.646, $p < 0.01$; r = 0.711, $p < 0.01$; and r = 0.332, $p < 0.01$. The perceived control dimension was significantly correlated with the two sub-dimensions of safety performance, predictive safety and perceived safety, and their correlation coefficients were, respectively, r = 0.212, $p < 0.05$ and r = 0.327, $p < 0.01$. According to the above-mentioned analysis results, it can be concluded in this study that the perceived control, safety attitude, and safety performance of the subjects were closely and significantly correlated with one another. Therefore, a hierarchical regression analysis was conducted to

further explore the interconnections among the subjects' perceived control, safety attitude, and safety performance.

**Table 6.** Pearson correlation analysis.

| | | (1) | (2) | (3) | (4) | (5) | (6) | (7) | (8) | (9) |
|---|---|---|---|---|---|---|---|---|---|---|
| (1) | Age | 1 | | | | | | | | |
| (2) | Total Years of Working Experience | 0.761 ** | 1 | | | | | | | |
| (3) | Perceived Control | −0.068 | −0.067 | 1 | | | | | | |
| (4) | Personal Safety | −0.073 | −0.034 | 0.217 * | 1 | | | | | |
| (5) | Co-worker Safety | −0.207 * | −0.140 | 0.395 ** | 0.463 ** | 1 | | | | |
| (6) | Supervisor Support | −0.263 ** | −0.284 ** | 0.252 ** | 0.300 ** | 0.629 ** | 1 | | | |
| (7) | Safety Understanding | 0.038 | 0.033 | 0.254 ** | 0.219 * | 0.380 ** | 0.488 ** | 1 | | |
| (8) | Predictive safety | −0.206 * | −0.145 | 0.212 * | 0.093 | −0.025 | 0.014 | −0.041 | 1 | |
| (9) | Perceived safety | −0.289 ** | −0.269 ** | 0.327 ** | 0.336 ** | 0.646 ** | 0.711 ** | 0.332 ** | −0.057 | 1 |

Note: * significant at $p < 0.05$; ** significant at $p < 0.01$.

*4.3. Hierarchical Regression Analysis*

The correlation analysis of this study only examined the correlations between two variables of the dimensions and sub-dimensions. Further exploration would consider if there was any interconnection, such as a mediation effect among the three dimensions of perceived control, safety attitude, and safety performance. Mediation effect refers to the effect of a variable or variables, known as mediator variable(s), that mediate the relationship between the independent variable(s) and the dependent variable(s). In the verification of a mediation effect, the theory proposed by Baron & Kenny [56] was used in this study. According to their theory, the following three conditions must be met to establish the existence of full mediation effect: (1) the independent variable has a significant influence on the mediator variable; (2) the independent variable has a significant influence on the dependent variable; and (3) the influence of the mediator variable reaches the level of significance, while the influence of the independent variable is insignificant in a regression analysis of the influences of the independent and moderator variables on the dependent variable.

The results were shown in Table 7. According to Model 1, the regression model was supported (F = 23.637 **, $p < 0.01$), indicating that perceived control was capable of explaining the variances of safety attitude. The adjusted $R^2$ was 0.162, indicating the capability of explaining 16.2% of the variances of the dependent variable (safety attitude). The perceived control dimension reached the level of significance (β = 0.411 **), meeting the first condition of the Baron & Kenny model. Moreover, according to Mode 1 of Model 2, the regression model was supported (F = 6.006 *, $p < 0.05$), indicating perceived control was capable of explaining the variances of safety performance (β = 0.219 *). The adjusted R2 was 0.048, indicating the capability of explaining 4.8% of the variances of the dependent variable (safety performance). This finding met the second condition of the Baron & Kenny model. Lastly, according to Mode 2 of Model 2, the regression model was also supported (F = 33.848 **, $p < 0.01$), indicating that, after its introduction into the model, safety attitude had a stronger influence than perceived control on safety performance with an adjusted $R_2$ of 0.362, while the β coefficient of perceived control was reduced from the level of significance to the level of insignificance (from −0.219 * to −0.034). The mediation effect of safety attitude was significant (β = 0.624 **, $p < 0.01$). The above-mentioned analysis results completely met the three conditions proposed by Baron & Kenny, supporting H4 of this study that the safety attitude of WWTP workers has a mediation effect between their perceived control and safety performance. It was also found that perceived control did not have a positive influence on safety performance but an indirect influence on safety performance via safety attitude.

**Table 7.** Hierarchical regression analysis results.

| | Model 1 | | | Model 2 Safety Performance | | | | | |
| | Safety Attitude | | | Mode 1 | | | Mode 2 | | |
| | Standardized β | t | VIF | Standardized β | t | VIF | Standardized β | t | VIF |
| --- | --- | --- | --- | --- | --- | --- | --- | --- | --- |
| Perceived Control | 0.411 | 4.862 ** | 1.000 | 0.219 | 2.451 * | 1.000 | 0.034 | 0.425 | 1.198 |
| Safety Attitude | - | - | - | - | - | - | 0.624 | 7.680 ** | 1.198 |
| $R^2$ | | 0.169 | | | 0.048 | | | 0.373 | |
| $Adj - R^2$ | | 0.162 | | | 0.040 | | | 0.362 | |
| F | | 23.637 ** | | | 6.006 ** | | | 33.848 ** | |
| DW | | 1.767 | | | 1.491 | | | 1.427 | |

Note: * significant at $p < 0.05$; ** significant at $p < 0.01$.

In addition to supporting H4, the analyses of this study also supported H1: "Perceived control of WWTP workers has a direct positive influence on their safety attitude" and H3: "Safety attitude of WWTP workers has a direct positive influence on their safety performance". However, H2 of this study, "Perceived control of WWTP workers has a direct positive influence on their safety performance", was not supported.

The coefficient of determination of this model 2 was 0.373, and only the path of safety attitude to safety performance reached the level of significance (=0.624, $p < 0.01$), indicating a direct and significant influence of safety attitude on safety performance. This finding also proved the direct causality between the WWTP's safety attitude and safety performance. Moreover, from the analysis, it could be found that the indirect influence of perceived control on safety performance via safety attitude was equal to the path coefficient of perceived control to safety attitude (=0.411) multiplied by the path coefficient of safety attitude to safety performance (=0.624). Therefore, the mediation effect of perceived control on safety performance was 0.256. Therefore, in Model 1, for example, the scores actually represented high perceived control, which had a positive influence on safety attitude.

## 5. Conclusions

The descriptive analysis results of this study demonstrated that the WWTP workers generally had a mid to high level of perceived control and recognized the relevance of the personal participation in the work safety management of their organizations. In terms of safety attitude, the subjects in this study demonstrated positive safety understanding about supervisors' measures, such as making round checks and stopping dangerous behaviors. However, it was also found in this study that further communication with WWTP workers is required to address their concerns about the possible hindrance of work efficiency caused by safety procedures. In terms of safety performance, the subjects did not demonstrate high predictive safety due to relatively low recognition or confidence in the safety protection at their work sites at the WWTPs. According to the variance analysis results, younger subjects generally had better scores in safety performance than older subjects, while subjects employed by WWTP operation contractors had significantly better scores in safety performance than subjects employed by WWTPs (which are considered as governmental agencies). It is suggested that safety management strategies that allow more opportunities of employee participation should be adopted to enhance WWTP workers' recognition and confidence. In addition, older workers and workers who are civil servants employed by governmental agenices should recieve more safety training and reminding to prevent hazards caused by their negligence or carelessness.

According to the regression analysis results of this study, perceived control had a positive influence on safety performance via safety attitude. This finding of the study is highly consistent with the findings of the research by Kuo et al. [26] on the manufacturing industry in Taiwan. It is difficult to change the existing attitudes and beliefs of an adult using direct measures, such as persuasion. However, with organizational control measures, it is possible

to change people's thinking and believing by changing their behaviors first [57]. Working at WWTPs with many confined spaces can be risky. As illustrated in Figures 1 and 2, there were, in total, 59 deaths and 39 injuries of workers in the 35 occupational hazards within confined spaces during the past 10 years in Taiwan. Among the 35 hazards, the worst one took as many as five lives. According to the existing research [9,10], the reasons for such deaths are hypoxia, intoxication, and a lack of proper personal protection when workers were working in confined spaces or trying to rescue their colleagues trapped in confined spaces. Therefore, if workers can have better perceived safety and safety attitudes, it is possible to reduce both the possibility and casualty rate of occupational hazards.

The key to the prevention of man-made hazards lies in improving the awareness, knowledge, and ability of people to ensure better safety. As suggested by Oswald et al. [58], Zhou et al. [59], and Rodríguez-Garzón et al. [60], training can serve as a critical factor in the improvement of the safety climate, perceived safety, and safety behaviors [43]. Besides, Zohar [61] indicated that the successful improvement of the safety level in industrial organizations highly depends on both genuine changes in attitudes and higher levels of commitment from the management. Hinze et al. [62] also pointed out that support from top management is needed to ensure successful safety management. As human negligence is the major cause of many site accidents, management diligence and support are required to better prevent accidents. Therefore, building a comprehensive safety culture based on safety training and safety awareness promotion is an important goal that deserves persistent attention and effort. It is suggested that employers, managers, workers, and all the other interested parties communicate and share thoughts and measures about safety management not only at meetings but also in daily-life situations. Managers and supervisors are also suggested to lead by example, which will encourage their subordinates to follow suit and build a healthy safety culture at their companies or organizations. Hopefully, with everyone in the workplace taking work safety management seriously, the future of autonomous safety management will become a reality.

To minimize the influence of variances caused by frequent changes of the team composition of workers and changes of work sites, this study mainly focused on the workers responsible for WWTP operation and repair/maintenance employed by WWTPs and their contractors for they mostly work within the same teams at the same work sites. Things will be different for studies on workers of contractors/subcontractors for WWTPs under construction. Such contractors obtain their business through bidding, and their businesses tend not to be stable or large enough to support regular teams of many workers. As a result, each of their subcontractors mostly only has a team of a dozen workers or even less than 10 workers who are often assigned to different crews at different work sites. In addition, over 90% of these workers are daily workers, and they come and go frequently. Under such challenging circumstances, it requires different strategies to promote higher safety awareness and better safety performance among such short-term and irregular teams of workers. As indicated by the existing research, building a good organizational culture is helpful for improving the safety performance among employees. However, short-term workers may build a rapport with their crew members, but they do not necessarily identify with the contractors/subcontractors who hire them, let alone the WWTPs. For such workers, there are still several gaps of identification with the organizational cultures of the contractors/subcontractors and the WWTPs to overcome in order to ensure better safety attitudes and safety performance among them. It is a worthwhile topic for future research.

**Author Contributions:** Conceptualization, C.-F.H. and Y.-L.T.; methodology, C.-F.H. and Y.-L.T.; data curation, Y.-L.T. and W.-H.L.; formal analysis, C.-F.H. and W.-H.L.; investigation, Y.-L.T. and W.-H.L.; project administration, C.-F.H.; writing—original draft preparation, C.-F.H. and Y.-L.T.; writing—review and editing, C.-F.H. and W.-H.L. All authors have read and agreed to the published version of the manuscript.

**Funding:** This research received no external funding.

**Institutional Review Board Statement:** Not applicable.

**Informed Consent Statement:** Not applicable.

**Data Availability Statement:** The data presented in this study are available on request from the second author. The data are not publicly available due to privacy considerations.

**Acknowledgments:** The authors would like to express their sincere gratitude to the anonymous reviewers who significantly enhanced the contents of the study with their insightful comments.

**Conflicts of Interest:** The authors declare no conflict of interest.

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
