# Peer review of "Relationships among Perceived Control, Safety Attitude, and Safety Performance: A Case Study on Wastewater Treatment Plant Workers"

_sustainability, doi:10.3390/su132212573_

Round 1

Reviewer 1 Report

At page 1, last line, references should be added to "... have been reported".

At page 5, in the  middle, "Heinrch (26)" is cited. In the references it is Heinrich.

At page, H4, the formatting should be improved.

Author Response

First of all, thank you for your valuable inputs, which help to enrich and improve this study. The paper has been revised as much as possible based on your suggestions and opinions. The revised parts are indicated in red in the latest version. Hopefully, the revisions can answer the questions or concerns you have raised. The following are our responses to your opinions.

<Opinions of the First Reviewer and Reponses>

  1. Reviewer’s Opinion

At page 1, last line, references should be added to "... have been reported".

  1. Response

   Revised as suggested. Please see p. 1.

  1. Reviewer’s Opinion

At page 5, in the middle, "Heinrch (26)" is cited. In the references it is Heinrich.

  1. Response

   Revised as suggested. Please see p. 5.

  1. Reviewer’s Opinion

At page, H4, the formatting should be improved.

  1. Response

   Revised as suggested. Please the revision marked in red in section 2.1~2.4.

Reviewer 2 Report

The authors have conducted an interesting research on the relationships among perceived control, safety attitude and Performance on wastewater treatment plant workers. It is an important topic, the flow of the research is clear, and the manuscript is organized well. I recommend this manuscript for publication after some minor revisions. My comments are as follows.

  • Reduce the size of Figures.
  • Add a research flowchart in Section 3 to make it easier for the reader and see the steps of your research clearer.
  • Add the conceptual framework of your research.
  • Rename section 5 to “Conclusion”

Author Response

First of all, thank you for your valuable inputs, which help to enrich and improve this study. The paper has been revised as much as possible based on your suggestions and opinions. The revised parts are indicated in red in the latest version. Hopefully, the revisions can answer the questions or concerns you have raised. The following are our responses to your opinions.

<Opinions of the Second Reviewer and Reponses>

1. Reviewer’s Opinion

Reduce the size of Figures..

1. Response

 Revised as suggested. Please see p. 3.

2. Reviewer’s Opinion

Add a research flowchart in Section 3 to make it easier for the reader and see the steps of your research clearer.
2. Response
 The process flow of this study is not complicated. Considering the limited length of this paper, we still decide to explain the process in writing instead of adding a flowchart as suggested. Our explanation of the process is included in Section 3.2 “Questionnaire Design and Sampling Method” (renamed from “Questionnaire Design”) in p. 6-7.

3. Reviewer’s Opinion

Add the conceptual framework of your research.

3. Response

The conceptual framework of our research can be known from the hypothesis in Section 3.1, the revised Section 3.2, and the research findings in Section 4.3. Therefore, we believe that it is not necessary to add the conceptual framework.

4. Reviewer’s Opinion
Rename section 5 to “Conclusion”

4. Response

 Revised as suggested. Please see p. 15.

Reviewer 3 Report

Review for sustainability

General comments

I have reviewed your manuscript and would recommend that it is accepted subject to addressing the comments below. The topic is interesting, but the knowledge gap is not demonstrated. The results are informative but need to be discussed further and a conclusion section created. There are a few grammatical errors including tautology. 

Introduction 

The approach to introducing the work in turn attempting to make a case for the study is logical. You attempted to demonstrate the significance of the study covering many hazards in the activities or operations. However, more could have been done to make the case stronger just as the significance of the study. I believe that using statistics to support your points will make it stronger. Also, adopt the funnel approach in presenting this. 

While you covered safety self-control, safety awareness, safety attitude, observance of safety protocol of WWTP workers in such hazardous working environment in the literature review, you have not adequately demonstrated the connection to the paragraphs before the last one in the introduction. The first sentence in the last paragraph of the introduction is about the only attempt to connect this here. How have self-control, safety awareness, safety attitude, observance of safety protocol of WWTP workers in such hazardous working environment helped improve health and safety performance? You can use statistics to demonstrate this.  

The knowledge gap also needs to be demonstrated in this section. Alternatively, it can be outlined and demonstrated in the literature review.

Clearly state the objectives of the study. The aim is noted, well done.

Literature review 

2.1

In the last paragraph of 2.1, you noted ‘There are already studies that have found various influence factors of safety behaviors and outcomes [11]and explored the influence of organizational safety climate on safety behaviors and outcomes [12]. The influences of personal factors of WWTP workers, such as their perceived control and safety attitude, on their safety performance, are still understudied. Therefore, this study was conducted focusing on the personal psychological aspects of workers to provide helpful suggestions based on its findings for WWTPs’. However, you have not adequately shown the knowledge gap here. The above is not enough. You need to demonstrate this knowledge gap. 

2.2

You need to show how you arrived at the safety attitude indicators used in the study. Of course, it must be underpinned by literature. Attempting to do this in the literature review section is not enough. 

Research methods. 

The analytical framework of the study needs to be presented graphically. Here all the indicators will also be shown. Instead of listing the literature that informed the questionnaire here, why don’t you use a table and do this in the literature review section?   

Provide more contextual information about the four wastewater treatment plants where the data was connected.

 How were the questionnaires distributed? 

Results

These are informative 

Discussion 

This has been covered in the ‘conclusion and suggestion’ section. This is inadequate. Most of the content in the section ‘conclusion and suggestion’ should come under the discussion section which should be created. A new conclusion and recommendation section should then be created. Of course, the conclusion and recommendation section should start by reminding the readers of what you did before the major findings and implications are presented. No new points should be here. The recommendations for addressing the issues in the study and further research must be covered just as the limitations of the research. 

On a different note, I think the studies below will help improve the literature review or discussion.

Clarke, S., 2006. The relationship between safety climate and safety performance: a meta-analytic review. Journal of occupational health psychology, 11(4), pp. 315-327. 

Umar, T. and Umeokafor, N.2021. A new safety climate assessment tool for Gulf construction. In: Sandanayake, Y.G., Gunatilake, S. and Waidyasekara, KG.A.S. (eds). Proceedings of the 9th World Construction Symposium, 9-10 July 2021, Sri Lanka. [Online]. pp. 39-51. DOI: https://doi.org/10.31705/WCS.2021.4. 

Author Response

First of all, thank you for your valuable inputs, which help to enrich and improve this study. The paper has been revised as much as possible based on your suggestions and opinions. The revised parts are indicated in red in the latest version. Hopefully, the revisions can answer the questions or concerns you have raised. The following are our responses to your opinions.

<Opinions of the Third Reviewer and Reponses>

1. Reviewer’s Opinion

Introduction

 The approach to introducing the work in turn attempting to make a case for the study is logical. You attempted to demonstrate the significance of the study covering many hazards in the activities or operations. However, more could have been done to make the case stronger just as the significance of the study. I believe that using statistics to support your points will make it stronger. Also, adopt the funnel approach in presenting this.

1. Response

  Thank you for your suggestions. We have made quite a few revisions and additions (marked in red) in response to your suggestions in the hope of strengthening our narratives.

2. Reviewer’s Opinion
While you covered safety self-control, safety awareness, safety attitude, observance of safety protocol of WWTP workers in such hazardous working environment in the literature review, you have not adequately demonstrated the connection to the paragraphs before the last one in the introduction. The first sentence in the last paragraph of the introduction is about the only attempt to connect this here. How have self-control, safety awareness, safety attitude, observance of safety protocol of WWTP workers in such hazardous working environment helped improve health and safety performance? You can use statistics to demonstrate this.

2. Response
(1) The suggested connection has been added in sections 2.1 to 2.4.
(2) There are limited statistics from research related to WWTP. The related statistics we have found so far are included in Figure 1 and Figure 2.

3. Reviewer’s Opinion
The knowledge gap also needs to be demonstrated in this section. Alternatively, it can be outlined and demonstrated in the literature review.

3. Response
The suggested connection has been added in the Introduction and in Sections 2.1 to 2.4.

4. Reviewer’s Opinion
Literature review

2.1

  In the last paragraph of 2.1, you noted ‘There are already studies that have found various influence factors of safety behaviors and outcomes [11]and explored the influence of organizational safety climate on safety behaviors and outcomes [12]. The influences of personal factors of WWTP workers, such as their perceived control and safety attitude, on their safety performance, are still understudied. Therefore, this study was conducted focusing on the personal psychological aspects of workers to provide helpful suggestions based on its findings for WWTPs’. However, you have not adequately shown the knowledge gap here. The above is not enough. You need to demonstrate this knowledge gap.

4. Response
Three more studies have been quoted in Section 2.1 to fill the knowledge gap. More papers have also been quoted in the Literature Review and following sections. Thank you for your helpful suggestion.

5. Reviewer’s Opinion
2.2

   You need to show how you arrived at the safety attitude indicators used in the study. Of course, it must be underpinned by literature. Attempting to do this in the literature review section is not enough.

5. Response
In the Literature review, 13 more studies have been quoted to improve the connection and fill the knowledge gap.

6. Reviewer’s Opinion
Research methods.

   The analytical framework of the study needs to be presented graphically. Here all the indicators will also be shown. Instead of listing the literature that informed the questionnaire here, why don’t you use a table and do this in the literature review section?  

   Provide more contextual information about the four wastewater treatment plants where the data was connected. How were the questionnaires distributed?

6. Response
(1) Thank you for your suggestion. Presentation of the analytical framework or literature review references through graphics or tables is one of the expression approaches. However, different journals and different writers have different approaches. The framework of this study can be known from the hypotheses in Section 3.1, the revised Section 3.2, and the research findings in Section 4.3. Given the limited length of this paper, we believe that graphic presentation of the framework is omissible.

(2) Further explanation about the questionnaire survey and sampling method has been added in the revised Section 3.2.

7. Reviewer’s Opinion
This has been covered in the ‘conclusion and suggestion’ section. This is inadequate. Most of the content in the section ‘conclusion and suggestion’ should come under the discussion section which should be created. A new conclusion and recommendation section should then be created. Of course, the conclusion and recommendation section should start by reminding the readers of what you did before the major findings and implications are presented. No new points should be here. The recommendations for addressing the issues in the study and further research must be covered just as the limitations of the research.

  On a different note, I think the studies below will help improve the literature review or discussion.
Clarke, S., 2006. The relationship between safety climate and safety performance: a meta-analytic review. Journal of occupational health psychology, 11(4), pp. 315-327.

Umar, T. and Umeokafor, N., 2021. A new safety climate assessment tool for Gulf construction. In: Sandanayake, Y.G., Gunatilake, S. and Waidyasekara, KG.A.S. (eds). Proceedings of the 9th World Construction Symposium, 9-10 July 2021, Sri Lanka. [Online]. pp. 39-51. DOI: https://doi.org/10.31705/WCS.2021.4.

7. Response
(1) Based on your opinions and those of the other reviewer, we have renamed the “Conclusion and Suggestion” section to “Conclusion”, in which we have also added our research restrictions and suggestions for future research. We believe that the revised “Conclusion” section has included all the suggested elements. Thank you for your suggestion and we will focus more on the presentation of research results in our future papers.

(2) We obtained and carefully read the two recommended papers, which led us to more related papers. We have included the two recommended papers and several of the related papers. Thank you for your suggestion and recommendation which have helped to enrich our paper.